# ADAPT: Adaptive Prompt Tuning for Pre-Trained Vision-Language Models

## Abstract

Prompt tuning has emerged as an effective way for parameter-efficient fine-tuning. Conventional deep prompt tuning inserts continuous prompts of a fixed context length into the input to each layer. When a pre-trained model is tailored to a specific downstream task, different layers initialized with pre-trained weights might have different levels of deviation from the optimal weights. Inserted prompts with a fixed context length might have redundant context tokens or insufficient context length. To address this issue, we propose a deep continuous prompting method dubbed Adapt that encourages heterogeneous context lengths. In this method, context lengths are automatically determined by iteratively pruning context tokens. We use the saliency criterion for neural network pruning to compute the importance scores of context tokens in order to determine which tokens to prune. To avoid the forgetting issue in the fine-tuning process, we apply the angular knowledge distillation to force the model to learn the angular separation between pairs of classes and that of instances from the pre-trained model. We examine the proposed method on the pre-trained vision-language model CLIP. 16-shot experiments on 11 downstream datasets reveal the advantage of Adapt: the average test accuracy achieves competitive performance, and the highest performance gain on individual datasets is 7.44%. We release the code in `https://anonymous.4open.science/r/Adapt-Prompt-Release`.

## 1 Introduction

Large-scale models have gained significant attention in language Brown (2020); Wang et al. (2021); Touvron et al. (2023), vision He et al. (2022); Zou et al. (2024); Esser et al. (2024) and multimodality Radford et al. (2021); Lu et al. (2022); Lai et al. (2024); Liu et al. (2024). When applying pre-trained large-scale models to various downstream tasks, the zero-shot performance can be sub-optimal. Although fine-tuning remarkably elicits the potential of pre-trained models, the fine-tuning process is computationally expensive. Parameter-efficient fine-tuning (PEFT) offers an efficient way to adapt the pre-trained model to various downstream tasks at a low cost. PEFT enhances the performance of pre-trained models by reparametrizing model weights Hu et al. (2021); Dettmers et al. (2024); Zhao et al. (2024), additive modules Chen et al. (2022b); Zhang et al. (2023); Mou et al. (2024) and selective weight updates Ding et al. (2023); Lawton et al. (2023); Fu et al. (2023). Among PEFT methods, prompting methods have the least effect on backbone models as they focus on the input of layers instead of model parameters. Prompt-based adaptation has been applied in various areas, including foundational computer vision (CV) tasks, constrained learning, trustworthy AI, foundational analysis, etc. Xiao et al. (2025b).

Prompts can be categorized into discrete prompts and continuous prompts. Discrete prompts use concrete word tokens to prompt pre-trained models. Compared to discrete prompts, continuous prompts (also called soft prompts) represent trainable tokens in a continuous embedding space. Hence, continuous prompts are differentiable and parameterized by their weights. They can be automatically tuned conditioning on downstream tasks.

Continuous prompts have shown competitive performance in language Li & Liang (2021); Gu et al. (2021); Liu et al. (2021; 2023), vision Jia et al. (2022); Bahng et al. (2022); Han et al. (2023) and multimodality

Zhou et al. (2022b); Shu et al. (2022); Ju et al. (2022); Wang et al. (2022). Existing continuous prompting methods use the prompt depth and context length to design continuous prompts. The underlying constraint is that the context length remains constant at different depths. However, if different layers have different levels of deviation from the optimal weights for downstream tasks, the constraint might be detrimental to the performance.

Recent works Lee et al. (2022); Chiatti et al. (2023); Panigrahi et al. (2023) have found that some layers of pre-trained models are closer to the optimal for downstream tasks. Fine-tuning layers that are far away from the optimal weights can achieve better performance than training all the layers uniformly. For prompting methods, we postulate that the layers far away from the optimal weights require longer context length while the layers close to the optimal weights demand shorter context length or even no context token. Hence, we seek to remove the constraint in the existing continuous prompting methods that require the same context length at different depths.

To this end, we propose a method dubbed **ada**ptive **p**rompt **t**uing (Adapt) that automatically determines context lengths at various depths. During the training process, Adapt uses time-varying binary masks to dynamically control context lengths. The variation of the binary mask depends on the importance of context tokens. The least important context token is constantly removed until the budget (a hyperparameter to control the total context length) is reached. We test the performance of Adapt on various downstream tasks and report a new state-of-the-art average accuracy. To our best knowledge, this is the first work to prune prompts for achieving heterogeneous context lengths in prompt learning.

The main contributions of Adapt are summarized below:

- We propose a method that removes the constraint in the existing continuous prompting methods that context lengths remain constant through the entire prompt depth. Adapt encourages a more flexible and optimal design for prompting methods.

- Context lengths are automatically determined in a non-parametric manner: prompts are initialized with the maximum context length and then iteratively pruned based on the importance score of context tokens. We use saliency criteria to characterize the importance of inserted context tokens. Pruning can effectively reduce the computational overhead with minimal performance drop.

- Context lengths can vary based on the downstream datasets. We use a hyperparameter of total context length to ensure the complexity of Adapt on various datasets is approximately the same while being heterogeneous.

- We apply angular knowledge distillation by matching the angular separation between pairs of images (image branch) and classes (text branch) for improvement.

## 2 Related Work

**Prompt Tuning** Prompt tuning (PT) uses continuous prompts to improve the performance of pre-trained models in diverse downstream tasks. CoOp Zhou et al. (2022b) is the pioneering work to apply PT for vision-language models. PT has shown great potential in various areas including image classification Zhou et al. (2022b;a); Hirohashi et al. (2024), out-of-distribution detection Miyai et al. (2024); Li et al. (2024), video understanding Ju et al. (2022); Huang et al. (2023), object detection Du et al. (2022); He et al. (2023), etc. Due to the good alignment of text and image representations of foundational vision-language models, there are emerging researches on applying those models such as CLIP Radford et al. (2021) to vision-language tasks. VPT Jia et al. (2022) proposes a paradigm of deep continuous prompting. PLOT Chen et al. (2022a) applies the optimal transport theory to improve the alignment between visual features and prompts. ProGrad Zhu et al. (2023) distills the prior knowledge from the pre-trained model to avoid forgetting issues Li & Hoiem (2017); Gou et al. (2021). MaPLe Khattak et al. (2023a) uses linear transformation layers to enhance the coupling between the text and image branches while UPT Zang et al. (2022) utilizes transformer blocks. PromptSRC Khattak et al. (2023b) applies the self-regulation strategy to avoid the forgetting issue. DAPT Cho et al. (2023) proposes the distribution-aware prompts. LAMM Gao et al. (2024) uses dynamic

category embedding and hierarchical loss to achieve an appropriate label distribution. ViAPT proposes instance-aware prompts to balance dataset-level and instance-level knowledge Xiao et al. (2025a).

**Network Pruning** Over-parametrization is a well-known property of deep neural networks. Network pruning removes unimportant model parameters to improve efficiency. It can be categorized into structured pruning and unstructured pruning. Unstructured pruning such as Han et al. (2015) removes individual parameters while structured pruning such as Liu et al. (2018) prunes models at a higher level (*e.g.* neurons, filters, and layers). A fundamental question in network pruning is to identify a saliency criterion to determine the importance of model parameters. Snip Lee et al. (2018) is a classic way to characterize the importance and can lead to a very sparse network without sacrificing too much performance.

## 3 Adaptive Prompt Tuning (Adapt)

We examine Adapt on the vision-language model CLIP Radford et al. (2021). CLIP is pre-trained over 400 million image-text pairs. The pre-training process is in a contrastive learning fashion to promote the alignment between text and image representations. CLIP consists of an image encoder and a text encoder. The prediction is done by matching the text and image representations.

### 3.1 Revisiting CLIP

Given an input image $\mathbf{I} \in \mathbb{R}^{H \times W \times 3}$, the image encoder splits it into fixed-size patches that are projected into patch embeddings $\mathbf{x} \in \mathbb{R}^{(N_i-1) \times d_i}$ Dosovitskiy (2020). $d_i$ is the hidden dimension on the image branch and $N_i$ is the input sequence length. A learnable classification token embedding $\mathbf{c}_i^{(0)}$ is prepended to the patch embeddings. The concatenated sequence of embeddings is passed to $\ell$ transformer blocks:

$$[\mathbf{c}_i^{(l)}, \mathbf{E}_i^{(l)}] = f^{(l)}([\mathbf{c}_i^{(l-1)}, \mathbf{E}_i^{(l-1)}]) , \tag{1}$$

where $l \in \mathbb{N}^+, 1 \leq l \leq \ell$, $f^{(l)}$ is the $l$-th transformer block of the image encoder. $\mathbf{E}_i^{(0)} = \mathbf{x}$. In the head of the image encoder, a linear transformation layer $\pi_i : \mathbb{R}^{d_i} \to \mathbb{R}^d$ transforms the classification token embedding in the image branch to the image representation $\mathbf{f}$.

A text prompt is fed to the text encoder to obtain the text embedding $\mathbf{E}_t = [\mathbf{w}^1, \mathbf{w}^2, \ldots, \mathbf{w}^{N_t}] \in \mathbb{R}^{N_t \times d_t}$. $d_t$ is the hidden dimension on the text branch. The text embedding contains the classification token embedding as the first token embedding. The text embedding is passed to $\ell$ transformer blocks:

$$\mathbf{E}_t^{(l)} = g^{(l)}(\mathbf{E}_t^{(l-1)}) , \tag{2}$$

where $g^{(l)}$ is the $l$-th transformer block of the text encoder. In the head of the text encoder, a linear transformation layer $\pi_t : \mathbb{R}^{d_t} \to \mathbb{R}^d$ transforms the classification token embedding in the text branch to the text representation $\mathbf{g}$.

The prediction for the input image $\mathbf{I}$ is computed by the cosine similarity between the text embedding and the image embedding:

$$p(y = k | \mathbf{x}) = \frac{\exp(\cos(\mathbf{f}_k, \mathbf{g})/\tau)}{\sum_{m=1}^{K} \exp(\cos(\mathbf{f}_m, \mathbf{g})/\tau)} . \tag{3}$$

Here $\tau$ is the temperature parameter, $K$ is the total number of classes.

### 3.2 Dynamic and Heterogeneous Soft Prompts

Figure 1 (a) showcases the traditional shallow and deep prompts for vision-language models. Figure 1 (b)-(e) show the proposed Adapt method. Different from the traditional deep prompting method, the Adapt method enables heterogeneous prompts inserted without changing the context lengths of hidden representations.

In the fine-tuning process of the pre-trained model, Adapt maximizes the likelihood of the correct label $y$:

$$\max_{\mathbf{P} \odot \mathcal{M}(t)} \mathbb{P}_{\mathbf{P} \odot \mathcal{M}(t), \boldsymbol{\theta}}(y | \mathbf{x}, \mathbf{P} \odot \mathcal{M}(t), \boldsymbol{\theta}) , \tag{4}$$

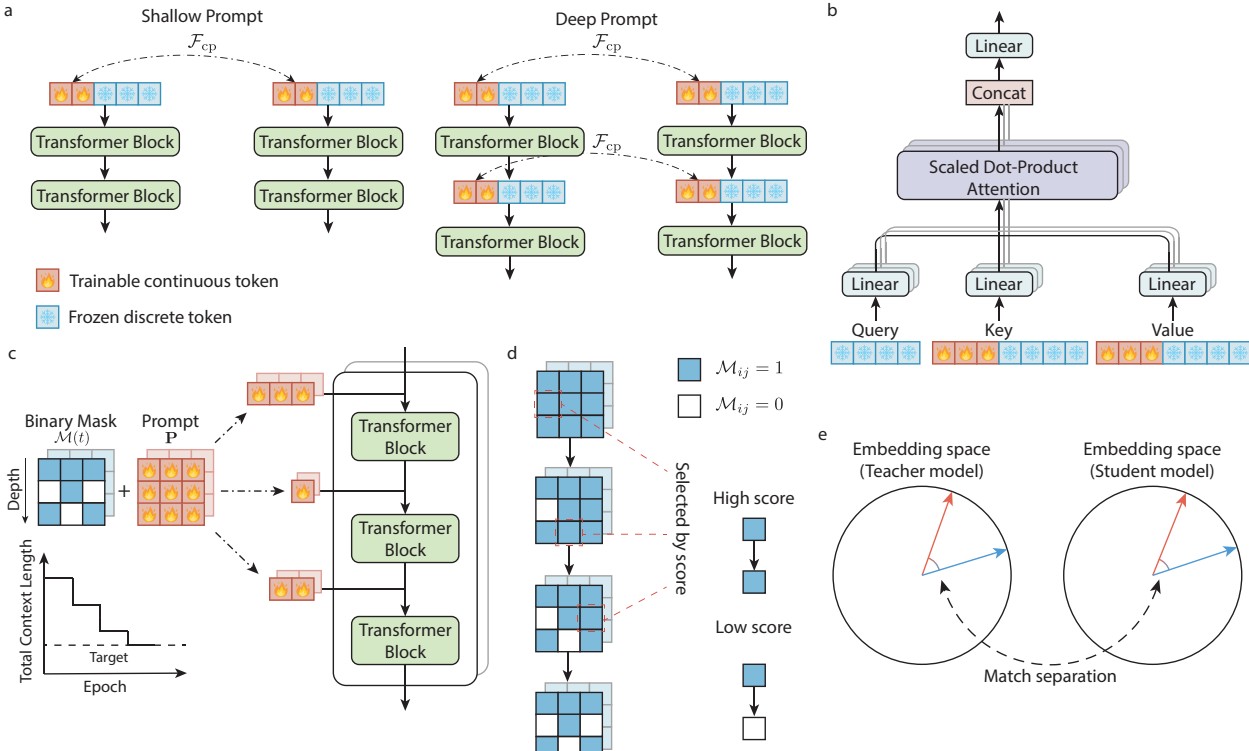

Figure 1: **Overview**. (a) The prompting paradigm of existing shallow and deep prompting methods. A coupling function $\mathcal{F}_{cp}$ can be applied to bridge the text branch to the image branch. (b) In the Adapt framework, we insert continuous prompts for key and value computation. The backbone model is frozen during the fine-tuning process. Only continuous prompts are trainable. (c) The proposed Adapt method encourages the pre-trained model to insert prompts with different context lengths. We use two binary masks $\mathcal{M}_f(t)$ and $\mathcal{M}_g(t)$ to adaptively control context lengths. Context lengths constantly change until the target $\mathcal{T}_{\text{target}}$ is reached. (d) The selection of context tokens to be pruned is based on the saliency scores. (e) To avoid the forgetting issue in the fine-tuning process, our method distills prior knowledge from the pre-trained model by learning the angular separation between the student model and the teacher model.

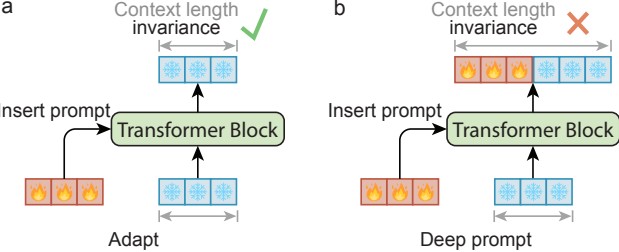

Figure 2: Comparison between (a) the Adapt method and (b) deep prompting paradigm for vision-language models. Adapt does not change the sequence length and hence enables heterogeneous context lengths at various depths, *i.e.*, context lengths are different at different depths.

where $\boldsymbol{\theta}$ is the weight of the pre-trained model that is frozen during the fine-tuning process. $\mathbf{P} \in \mathbb{R}^{\ell \times \xi \times d}$ is the inserted continuous prompt. $\xi$ is the maximum context length at various depths. $\mathcal{M}(t) \in \{0,1\}^{\ell \times \xi}$ is a time-dependent (dynamic) binary mask. We use $\odot$ to denote a modified Hadamard operation $\mathbf{M} = \mathbf{P} \odot \mathcal{M}(t)$, where $M_{ijk} = P_{ijk} \mathcal{M}(t)_{ij}, 1 \leq i \leq \ell, 1 \leq j \leq \xi, 1 \leq k \leq d$. For the vision-language model, there are two sets of independent prompts and binary masks. The optimization objective is over $\mathbf{P}_f \odot \mathcal{M}_f(t)$ and $\mathbf{P}_g \odot \mathcal{M}_g(t)$. $\mathbf{P}_f$ and $\mathbf{P}_g$ are prompts for image and text branches. $\mathcal{M}_f(t)$ and $\mathcal{M}_g(t)$ are binary masks for image and text branches.

We describe the optimization process of Adapt for the vision-language model as:

$$
\underset{\substack{\mathbf{P}_f, \mathcal{M}_f(t), \\ \mathbf{P}_g, \mathcal{M}_g(t)}}{\operatorname{argmin}} \quad \frac{1}{|\mathcal{D}|} \sum_{\mathbf{x}, y \in \mathcal{D}} \mathcal{L}(\mathbf{x}, y | \mathbf{P}_f, \mathcal{M}_f(t), \mathbf{P}_g, \mathbf{M}_g(t), \boldsymbol{\theta}) ,
$$

$$
\text{s.t.} \quad \sum_{i=1}^{\ell_f} \sum_{j=1}^{\xi_f} \mathcal{M}_f(t)_{ij} + \sum_{i=1}^{\ell_g} \sum_{i=1}^{\xi_g} \mathcal{M}_g(t)_{ij} \leq \mathcal{T}_{\text{target}} ,
\tag{5}
$$

where the hyperparameter $\mathcal{T}_{\text{target}}$ is the target total context length. It determines the complexity of the Adapt method. For brevity, we do not explicitly mention $\mathcal{M}_f(t)$ for the image branch and $\mathcal{M}_g(t)$ for the text branch. Instead we use $\mathcal{M}(t)$ as it can be applied to both the image and text branch. $\mathcal{M}(t)$ is initialized to be $\mathcal{M}(0) = \mathbf{1}_{\ell \times \xi}$ (an all-one matrix). At each iteration, we identify which token to prune and set the corresponding binary mask to be 0, *i.e.* $\mathcal{M}(t)_{ij} = 0$. The total context length continuously decreases until $\mathcal{T}_{\text{target}}$ is reached. We use $\mathcal{T}_{\text{target}} \ll \ell \times \xi$ to ensure the efficiency of the Adapt method.

In the pruning process, which context token to prune, *i.e.* finding $i, j$ and set $\mathcal{M}(t)_{ij} = 0$, is determined by the importance of corresponding context tokens as shown in Figure 1 (d). We borrow the saliency criterion widely used in the unstructured network pruning literature to measure the importance of context tokens. Specifically, we use Snip Lee et al. (2018), gradient norm and $l_2$-norm to compute the importance scores $S_c$ (also called saliency scores) for characterizing the importance. For the $t$-th context token ($t \in \mathbb{N}^+, 1 \leq t \leq \xi$) at the depth $l$ ($l \in \mathbb{N}^+, 1 \leq l \leq \ell$), the importance score computed by these three metrics is:

$$
\text{Snip: } S_c = \left| \frac{\partial \mathcal{L}}{\partial \mathbf{P}_{lt}} \odot \mathbf{P}_{lt} \right| ,
$$

$$
\text{gradient norm: } S_c = \left| \frac{\partial \mathcal{L}}{\partial \mathbf{P}_{lt}} \right| ,
\tag{6}
$$

$$
l_2\text{-norm: } S_c = |\mathbf{P}_{lt}| .
$$

$\mathcal{M}(t)$ controls the context length for each transformer block as shown in Figure 1 (b). $\mathcal{M}(t) \odot \mathbf{P}$ is the continuous prompt inserted to the pre-trained model. There is no constraint for context lengths at various depths. Hence, the added prompt $\mathbf{P} \odot \mathcal{M}(t)$ can be heterogeneous.

### 3.3 Prompt Tuning

Owing to $\mathcal{M}(t)$, the context length $\xi_l$ varies during the fine-tuning process. Unlike the existing deep prompting methods for the vision-language models that insert continuous prompts in the computation of key, value and query, Adapt inserts continuous prompts only for query and value in the self-attention Vaswani et al. (2017) as shown in Figure 1 (c). Different from the deep prompting paradigm, the context length does not change as indicated in Figure 2. Given an input $\mathbf{x}$ for a transformer block, the self-attention with inserted prompts in Adapt is computed by:

$$
\mathbf{Q} = f_q(\mathbf{x}) \in \mathbb{R}^{\xi_{\text{org}} \times d},
$$
$$
\mathbf{K} = f_k([\mathbf{P}, \mathbf{x}]) \in \mathbb{R}^{(\xi_{\text{org}} + \xi) \times d},
\tag{7}
$$
$$
\mathbf{V} = f_v([\mathbf{P}, \mathbf{x}]) \in \mathbb{R}^{(\xi_{\text{org}} + \xi) \times d} .
$$

$$
\text{Self-Attention} = \text{Softmax}(\frac{\mathbf{Q}\mathbf{K}^T}{\sqrt{d}})\mathbf{V} .
\tag{8}
$$

Here $f_q$, $f_k$ and $f_v$ are linear transformation functions for the query, key and value. $\xi_{\text{org}}$ is the sequence length of the input without the inserted prompt. During the fine-tuning process, the pre-trained model is frozen. Only inserted continuous prompts are optimized.

The proposed Adapt method for vision-language models is summarized in Algorithm 1. In the pruning step, the ranking is done based on scores in the image and text branches. The context tokens with the lowest score will be removed. The total context length in the text branch can be different from that in the image

branch. For the same branch, context lengths might vary at different depths. Hence, compared to the manually designed continuous prompt, Adapt can have highly heterogeneous context lengths. Besides, using the saliency criterion enables varying context lengths without additional trainable parameters.

Till the total context lengths of text and image branches reach $\mathcal{T}_{\text{target}}$, Adapt continues to remove context tokens. The accumulation period $n_k$ determines the number of accumulated steps to compute the score. The pruning rate $r_p$ dictates the number of removed tokens per pruning step.

---

**Algorithm 1** Adapt for vision-language models.

---

1: **Input**: A pre-trained vision-language model, prompt depth $\ell_f$ for the image encoder and $\ell_g$ for the text encoder, maximum context length $\xi_f$ for the image encoder and $\xi_g$ for the text encoder, target $\mathcal{T}_{\text{target}}$, accumulation period $n_k$ and pruning rate $r_p$.
2: Create a randomly initialized prompt $\mathbf{P}_f \in \mathbb{R}^{\ell_f \times \xi_f \times d}$ for the image branch and $\mathbf{P}_g \in \mathbb{R}^{\ell_g \times \xi_g \times d}$ for the text branch, and a binary mask $\mathcal{M}_f(0) = \mathbf{1}_{\ell_f \times \xi_f}$ for the image branch and $\mathcal{M}_g(0) = \mathbf{1}_{\ell_g \times \xi_g}$ for the text branch.
3: Initialize accumulated score $\mathbf{S}_f = \mathbf{0}_{\ell_f \times \xi_f}$ and $\mathbf{S}_g = \mathbf{0}_{\ell_g \times \xi_g}$.
4: **for** $t = \{1, \ldots, n_t\}$ **do**
5:     Insert the prompt $\mathbf{P}_f \odot \mathcal{M}_f(t)$ for the image branch and $\mathbf{P}_g \odot \mathcal{M}_g(t)$ for the text branch of the pre-trained model as shown in Equation 7 and 8.
6:     Perform forward and backward propagation using loss function in Equation 10 to update $\mathbf{P}_f$ and $\mathbf{P}_g$.
7:     **if** $\sum_{i=1}^{\ell_f} \sum_{j=1}^{\xi_f} \mathcal{M}_f(t)_{ij} + \sum_{i=1}^{\ell_g} \sum_{j=1}^{\xi_g} \mathcal{M}_g(t)_{ij} > \mathcal{T}_{\text{target}}$ **then**
8:         $\Delta\mathbf{S}_f \in \mathbb{R}^{\ell_f \times \xi_f}$ and $\Delta\mathbf{S}_g \in \mathbb{R}^{\ell_g \times \xi_g}$ according to Equation 6.
9:         $\mathbf{S}_f$ by $\mathbf{S}_f = \mathbf{S}_f + \Delta\mathbf{S}_f$ and $\mathbf{S}_g$ by $\mathbf{S}_g = \mathbf{S}_g + \Delta\mathbf{S}_g$.
10:         **if** $t == an_k, a \in \mathbb{N}^+$ **then**
11:             **for** Prune step $= \{1, \ldots, r_p\}$ **do**
12:                 Find location $(k_{\min}, i_{\min}, j_{\min}) = \text{argmin}_{k,i,j}\{[\mathbf{S}_k]_{ij}|\mathcal{M}_k(t)_{ij} == 1\}$.
13:                 $\mathcal{M}_{k_{\min}}(t)_{i_{\min}j_{\min}} = 0$.
14:             **end for**
15:             Reset accumulated score $\mathbf{S}_f = \mathbf{0}_{\ell_f \times \xi_f}$ and $\mathbf{S}_g = \mathbf{0}_{\ell_g \times \xi_g}$.
16:         **end if**
17:     **end if**
18: **end for**

---

### 3.4 Knowledge Distillation

Fine-tuning pre-trained models in various downstream tasks is subject to the forgetting issue Niu et al. (2022); Smith et al. (2023); Wang et al. (2022). Knowledge distillation Hinton et al. (2015) is an effective way to distill knowledge from the teacher model to the student model. A direct idea is to use the pre-trained model as the teacher model to distill the prior knowledge and to avoid the forgetting issue. However, there is a large performance gap between the pre-trained model and prompting methods as shown in Table 1. Hence, the logits predicted by the teacher model are sub-optimal. Based on PromptSRC Khattak et al. (2023b), where $l_1$ norm difference between embeddings of student model and teacher model is used as the knowledge distillation loss, and relational knowledge distillation Park et al. (2019), where similarity is used as the knowledge distillation loss, we propose angular knowledge distillation, which aligns the cosine similarity between embeddings of the student model and the teacher model. In vision-language models, the prediction is based on cosine similarity between text and image embeddings. Angular knowledge distillation considers cosine similarity between embeddings of the student model and the teacher model, which preserves the prior knowledge in the teacher model. The student model is forced to learn the angular separation between pairs of text embeddings for different classes and that between pairs of image embeddings for different images. Specifically, we apply a loss term associated with the angular separation:

$$\mathcal{L}_{\text{ang}} = \mathbb{E}_{\mathbf{x}^s \sim \mathcal{X}^s, \mathbf{x}^t \sim \mathcal{X}^t} \left| \langle \mathbf{x}_i^t, \mathbf{x}_j^t \rangle - \langle \mathbf{x}_i^s, \mathbf{x}_j^s \rangle \right| , \tag{9}$$

where $\langle \cdot \rangle$ computes the cosine similarity between two embeddings. $\mathbf{x}^s$ is the embedding generated by the student model while $\mathbf{x}^t$ is the embedding generated by the teacher model. $\mathbf{x}_i, \mathbf{x}_j \in \mathcal{X}$ are any pair of embeddings.

It has been found that a slight variation in the embedding can lead to classification error for vision-language models Cho et al. (2023); Hirohashi et al. (2024). $\mathcal{L}_{\text{ang}}$ helps the student model learn the prior knowledge by learning the angular separation between pairs of text embeddings of different classes and that of pairs of different image embeddings. The total training loss is computed by:

$$\mathcal{L}_{\text{total}} = \mathcal{L}_{\text{ce}} + \alpha \cdot \mathcal{L}_{\text{ang}}^{\text{text}} + \beta \cdot \mathcal{L}_{\text{ang}}^{\text{image}} . \tag{10}$$

Here $\alpha$ and $\beta$ are hyperparameters to control the contribution of two loss terms related to angular knowledge distillations. $\mathcal{L}_{\text{ce}}$ is the cross-entropy loss.

## 4 Experiments and Results

### 4.1 Experiments

**Datasets**   We examine the proposed Adapt method over 11 datasets: Caltech101 Fei-Fei et al. (2004) and ImageNet Deng et al. (2009) for the generic object recognition, Describable Textures Cimpoi et al. (2014) for the texture recognition, EuroSAT Helber et al. (2019) for the satellite image recognition, FGVCAircraft Maji et al. (2013), Food101 Bossard et al. (2014), OxfordFlowers Nilsback & Zisserman (2008), OxfordPets Parkhi et al. (2012), and StanfordCars Krause et al. (2013) for the fine-grained image recognition, UCF101 Soomro et al. (2012) for the action recognition, and SUN397 Xiao et al. (2010) for the scene recognition. We follow the few-shot learning setting in CoOp Zhou et al. (2022b). The number of shots is 16. For each dataset, the result is averaged over 3 runs. A detailed description of 11 datasets can be found in Appendix Section A.3.

**Baselines**   We compare the proposed method with CoOp Zhou et al. (2022b), VPT Jia et al. (2022), CoCoOp Zhou et al. (2022a), PLOT Chen et al. (2022a), UPT Zang et al. (2022), ProGrad Zhu et al. (2023), DAPT Cho et al. (2023), MaPLe Khattak et al. (2023a), PromptSRC Khattak et al. (2023b) and LAMM Gao et al. (2024). CoOp, CoCoOp, PLOT, ProGrad, LAMM and DAPT use shallow prompts while UPT, VPT-Deep, MaPLe and PromptSRC use deep prompts. CoOp Zhou et al. (2022b) inserts continuous prompts into the text prompt for vision-language models. CoCoOp Zhou et al. (2022a) adds continuous prompts conditioned on the input Image. VPT Jia et al. (2022) proposes a paradigm of the deep prompting method. ProGrad Zhu et al. (2023) proposes the gradient-aligned knowledge distillation. PLOT Chen et al. (2022a) utilizes the optimal transport theory Cuturi (2013) to align text and image embeddings. DAPT Cho et al. (2023) proposes distribution-aware prompting methods. UPT Zang et al. (2022) uses the transformer layer to contextualize prompt tokens in both text and image branches. MaPLe Khattak et al. (2023a) uses a linear projection in lieu of transformer layers to bridge the image branch and text branch. PromptSRC Khattak et al. (2023b) proposes self-ensembling for prompt templates, self-regulation for embeddings and prediction logits, and Gaussian weighted sampling for model weights. LAMM Gao et al. (2024) replaces the category tokens with trainable vectors and utilizes the hierarchical loss to preserve the generalization ability of the pre-trained model.

**Implementation Details**   We use the pretrained ViT-B/16 CLIP model Radford et al. (2021) in this work. The text template used for prompting the vision-language model is summarized in Appendix Section A.3. The number of mini-batches used for computing the score is $n_k = 80$. We use $\ell_f = \ell_g = 12$, $\xi_f = \xi_g = 18$ and $\mathcal{T}_{\text{target}} = 256$. In each pruning step, the number of pruned context tokens is $r_p = 2$. The batch size is 4. The learning rate is $2.5 \times 10^{-3}$. The total number of training epochs is 100. The test accuracy is obtained using the model weights at the epoch of 100. We use the stochastic gradient descent (SGD) to optimize the inserted prompts. Experiments are conducted using a single NVIDIA A40 GPU. Reported results on 11 datasets are averaged over 3 runs.

Table 1: Test accuracy comparison on various downstream tasks in the few-shot learning setting. The number of shots is 16. The reported performance is averaged over 3 runs. Adapt uses Snip to compute scores of context tokens.

| Method | Caltech101 | DTD | EuroSAT | Aircraft | Food101 | Flowers | Pets | Cars | Sun | UCF | ImageNet | Average |
|---|---|---|---|---|---|---|---|---|---|---|---|---|
| ZS CLIP | 87.20 | 42.34 | 37.57 | 17.29 | 77.30 | 66.18 | 85.79 | 55.63 | 58.55 | 61.45 | 58.20 | 58.86 |
| Linear Probe | 95.50 | 69.69 | 87.21 | 45.39 | 83.11 | 97.47 | 86.40 | 80.69 | 82.32 | 72.95 | 67.46 | 78.93 |
| VPT-Shallow | 94.66 | 52.27 | 84.93 | 30.9 | 86.91 | 81.42 | 92.61 | 69.05 | 75.18 | 68.07 | 68.98 | 73.18 |
| CoCoOp | 95.18 | 62.98 | 73.28 | 31.25 | 87.17 | 87.83 | 93.18 | 71.55 | 78.08 | 72.12 | 70.81 | 74.86 |
| PLOT | 93.70 | 70.90 | 84.03 | 34.93 | 78.13 | 97.27 | 88.20 | 68.10 | 72.23 | 69.40 | 72.17 | 75.37 |
| ProGrad | 95.63 | 66.27 | 82.03 | 41.30 | 86.70 | 95.33 | 93.10 | 81.23 | 81.60 | 75.13 | 72.27 | 79.14 |
| VPT-Deep | 95.83 | 69.77 | 91.60 | 40.88 | 86.17 | 94.97 | 92.91 | 76.09 | 82.74 | 71.62 | 70.57 | 79.38 |
| CoOp | 95.58 | 69.81 | 84.96 | 43.43 | 84.14 | 97.05 | 91.87 | 83.05 | 82.16 | 74.64 | 71.81 | 79.86 |
| UPT | 95.95 | 70.61 | 90.77 | 46.57 | 86.82 | 97.40 | 92.77 | 84.17 | 83.87 | 75.91 | 72.65 | 81.59 |
| LAMM | **96.90** | 72.21 | 82.71 | 45.42 | 86.43 | 97.36 | 93.11 | 85.10 | 83.78 | 76.09 | 72.71 | 81.07 |
| DAPT | 95.80 | 71.55 | 92.58 | 46.21 | 86.59 | 97.04 | 92.28 | 82.99 | 84.48 | 75.96 | 72.26 | 81.61 |
| MaPLe | 95.97 | 71.34 | 92.35 | 48.31 | 85.28 | 96.97 | 92.97 | 83.41 | 84.99 | 75.61 | 72.27 | 81.77 |
| PromptSRC | 96.04 | 72.75 | 92.48 | 50.86 | **87.44** | 97.62 | **93.81** | 83.79 | 86.43 | **77.24** | 73.12 | 82.87 |
| Adapt | 96.57 | **74.07** | **93.53** | **58.30** | 86.67 | **98.43** | 92.67 | **85.97** | 86.43 | 77.17 | **73.20** | **83.91** |

## 4.2   Results

The effectiveness of the Adapt method is examined using the few-shot learning setting. We summarize the experimental results in Table 1. We use Snip to compute scores. The performance of gradient norm and $l_2$-norm is reported in Appendix Figure 13 (c). Overall, the Adapt method exhibits superior performance compared to baseline methods. The Adapt method improves the best-known performance (82.87% by PromptSRC) to 83.91% on the average of 11 datasets. The largest performance gain is 7.44% on the FGVC Aircraft dataset. The performance of using fewer number of shots is reported in Appendix Figure 8. We notice that when the number of shots is less than 8, there is a degradation of the advantage of the Adapt method compared to baseline methods. This might be caused by poor convergence of bi-level optimization when the data are very limited.

Notably, Adapt inserts continuous prompts only for the key and value computations while the prevalent deep prompting methods for vision-language models insert continuous prompts for query, key and value computations. Using the same context length, this approach can effectively decrease the model complexity. Besides, the continuous prompts are not added to the query, which does not change the context length after the attention computation. Therefore, Adapt enables the addition of prompts with heterogeneous context lengths to the pre-trained model.

Adapt remarkably improves the performance of zero-shot CLIP model as shown in Appendix Figure 12. On each of 11 datasets, the Adapt method pronouncedly improves the performance. The highest test accuracy gain is 55.96% on the EuroSAT dataset. Compared to other prompting methods, there is also a large performance gap for the zero-shot CLIP model as shown in Table 1. This seems to indicate that the pre-trained CLIP model is not an ideal candidate as the teacher's model for Kullback–Leibler (KL) divergence on the prediction of the probability distribution of various classes. The angular distillation distills the prior knowledge by learning the angular separation, which avoids the distillation by the predicted probability distribution from the pre-trained CLIP model.

We examine the prompts after pruning. An example result is shown in Figure 3. The full results are shown in Appendix Figure 15. Adapt enables automatically determining highly heterogeneous prompts. The context lengths at different depths and different branches can be different.

## 4.3   Ablation Study

**Angular knowledge distillation**   Angular knowledge distillation enables the distillation of prior knowledge learned from the pre-training process. The traditional way of knowledge distillation (KD) is KL divergence Hinton et al. (2015) of the predicted probability distribution between the student model and the teacher model. However, the pre-trained CLIP model has inferior performance as shown in Table 1

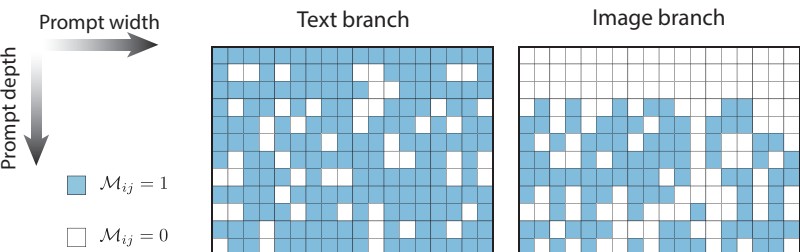

Figure 3: Example of prompts after pruning. The dataset is Stanford Cars. $\mathcal{M}_{ij} = 1$ indicates that the context token is used while $\mathcal{M}_{ij} = 0$ indicates that the context token is removed. Prompts after pruning are highly heterogeneous.

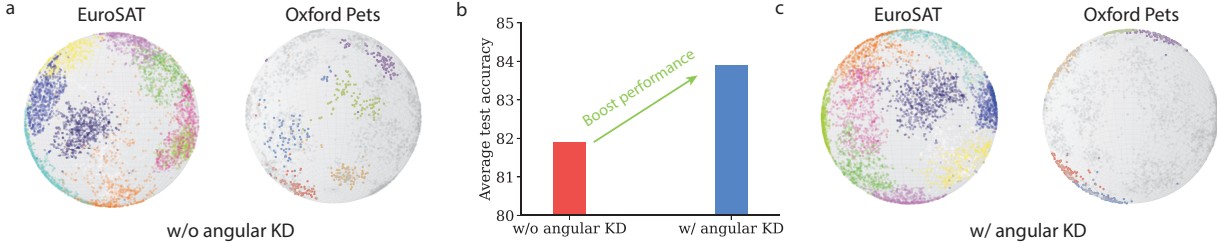

Figure 4: Effect of the angular knowledge distillation on model performance. (a) t-SNE Van der Maaten & Hinton (2008) image embeddings without applying angular knowledge distillation on downstream datasets. On the Oxford Pets dataset, we selectively highlight some classes for better visualization; (b) Applying angular knowledge distillation improves the average test accuracy; (c) t-SNE image embedding using angular knowledge distillation.

and Appendix Figure 12. The angular distillation avoids learning potentially sub-optimal prediction of the probability distribution.

Figure 4 (a) and (c) shows the distribution of image embeddings using angular KD and without using angular KD. Angular KD enables a better separation of embeddings of different classes and clustering of embeddings of the same classes. Figure 4 (b) shows the comparison of the average test accuracy over 11 datasets between these two approaches. The angular KD effectively improves the performance.

**Maximum context length** Considering the pruning process as a process of searching the optimal context length for each transformer block, the maximum context lengths $\xi_f$ for the image branch and $\xi_g$ for the text branch determine the size of the search space. When $\xi_f = \xi_g < 18$, the performance degrades even though $\mathcal{T}_{\text{target}}$ stays the same as shown in Figure 5. In other words, although the total context length of the final prompt stays the same, the maximum context length (essentially the size of the search space) has an effect on the performance of the Adapt method.

We notice that when $\xi_f$ and $\xi_g$ are small enough to enter the regime where the total context length before pruning is smaller than $\mathcal{T}_{\text{target}}$ (the shaded area in Figure 5), there is a remarkable performance degradation. The performance is even worse than using $\mathcal{T}_{\text{target}} = 128$. We use $\mathcal{T}_{\text{effective}}$ to represent the total context length after pruning. The red horizontal line in Figure 5 uses the $\xi_f = \xi_g = 18$ and $\mathcal{T}_{\text{target}} = 128$. The total context length after pruning is $\mathcal{T}_{\text{effective}} = \mathcal{T}_{\text{target}} = 128$. The performance is pronouncedly better than the two configurations in the shaded area. This indicates the pruning process not only decreases the computational cost but also improves the performance.

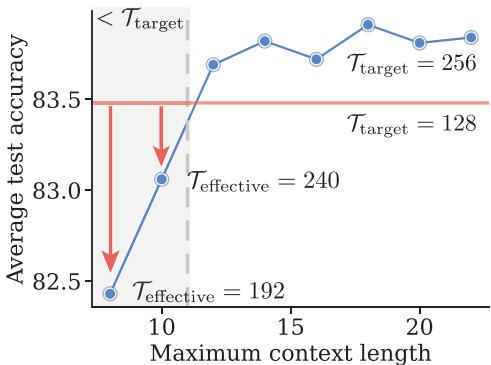

Figure 5: The effect of maximum context length $\xi_f = \xi_g$ on the model performance. In the shaded area, the total number of context tokens is smaller than $\mathcal{T}_{\text{target}}$ before pruning. Red line marks the performance of using $\mathcal{T}_{\text{target}} = 128$ in lieu of $\mathcal{T}_{\text{target}} = 256$. Even though the total context length (heterogeneous prompt) is smaller than that (homogeneous prompt without any pruning) in the shaded area, it has pronouncedly better performance.

## 5 Discussion

When tailoring a pre-trained model to various downstream tasks, the model can underperform Taori et al. (2020); Fang et al. (2020); Wiles et al. (2021); Xiao et al. (2024). When examining the model on a more granular level, a question arises "*is the inferior performance caused by the deviation from the optimal for all layers or a subset of layers*". Surgical fine-tuning Lee et al. (2022) finds that fine-tuning the selective part of the pre-trained model achieves a performance comparable to or better than training all layers. This result indicates that not all layers are at the same level of deviation from the optimal. Selectively training layers deviating from the optimal while keeping the remaining layers frozen achieves favorable performance.

In prompt tuning, the entire pre-trained model is frozen. Given the fact that some layers might already be near optimal, there is no need to insert continuous prompts for those layers. Prompts can be inserted into layers that are deviating from the optimal. If we consider this strategy in a more granular way, context lengths for different layers can vary depending on the level of deviation from the optimal. This leads to heterogeneous context lengths which are challenging for the manually designed prompting methods.

The proposed Adapt method achieves the automatic design of heterogeneous prompts. There is no constraint for context lengths at various depths to be the same, nor for context lengths to be the same for different branches. The results on 11 datasets indicate that context lengths can be highly heterogeneous as shown in Appendix Figure 15. The automation is achieved by iteratively pruning unimportant context tokens. By setting $\mathcal{T}_{\text{target}} \ll \ell_f \times \xi_f + \ell_g \times \xi_g$, the pruning greatly reduces the computational overheads. The comparison is based on Adapt before pruning and Adapt after pruning. The total number of trainable parameters is decreased by 41%. In the network pruning, pruning concentrated on one layer can cause the layer collapse issue Lee et al. (2019); Hayou et al. (2020). Pruning prompts, however, can have a minimal context length in one layer without affecting the functionality of prompts for this layer.

By using $\mathcal{M}(t)$ conditioning on downstream datasets, Adapt adaptively changes for different datasets. Compared to manually designed prompts, Adapt has a more flexible structure. It achieves a pronounced performance gain compared to baseline methods. We use $\mathcal{T}_{\text{target}}$ to control the complexity of Adapt.

## 6 Conclusion

We propose a continuous prompting method that adaptively changes during the fine-tuning process. Different from existing prompting methods that require homogeneous context lengths for various depths, our proposed method Adapt encourages heterogeneous context lengths, unlocking the space of prompt learning that was previously underexplored. Adapt uses iterative pruning to remove unimportant context tokens, which greatly reduces the computational costs. Compared to homogeneous prompting that is commonly

used in manually designed prompts, the heterogeneous prompting enabled by Adapt has better performance. Extensive experiments over 11 datasets exhibit the strength of the Adapt method, achieving new state-of-the-art performance in prompt learning for pre-trained vision-language models.

## 7 Broader Impact

We propose an automatic way to determine a heterogeneous prompt, which can be applied to fine-tune foundational models in various downstream tasks efficiently. Different from existing manually designed prompting approaches that use a homogeneous prompt, heterogeneous prompts improve efficiency by pruning unimportant context tokens. While this work focuses on VLMs, our method can, in principle, be applied to pure language and vision transformer models.

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
