# OpenReview forum: "ADAPT: Adaptive Prompt Tuning for Pre-Trained Vision-Language Models"
_TMLR — Rejected by TMLR_

### Review · Reviewer_hFh7 · 2025-10-27

**Summary Of Contributions:**

The paper proposes a set of methods, collectively called ADAPT which aim to improve performance in the context of parameter-efficient fine-tuning (PEFT) for Vision-Language Models. Specifically, two methods are evaluated in conjunction:

- A novel context-pruning mechanism based upon token saliency that is designed to converge on a set of added context tokens per Transformer block layer with varying cardinality,  in contrast to the prevailing approach of a fixed number of added tokens per layer.

- Angular knowledge distillation: An added regularisation term based on cosine similarity between the original teacher and new student model embeddings that is designed to penalise excessive divergence and hence preserve desirable generalisable properties of the original model, while facilitating fine tuning toward specific properties necessary for performance on new datasets.

Strengths
- The context-pruning mechanism is an original contribution with a reasonable motivation given for attempting it.
- The paper is clearly written and the various methods are generally laid out well.

Weaknesses
- The empirical results provided are weak and somewhat misleading in what is presented and what isn’t.
- Ablations that could support claims made are not provided.

**Audience:**

Yes

**Audience Explanation:**

If ADAPT was shown to be effective in materially reducing the number of new trainable parameters (while at least preserving accuracy) then this might be of interest to the community, particularly when considering very low resource settings or situations where performance on a new task was contingent on a substantial number of new trainable parameters being added.

**Broader Impact Concerns:**

I think this work is fine without an Impact Statement.

**Claims And Evidence:**

No

**Claims Explanation:**

The primary focus in the paper is the context pruning method based on saliency. Unfortunately, the results (Figure 4b) suggest that the main driver of improvement is actually the added regularisation term based on cosine similarity (angular knowledge distillation). This approach was one of several similar methods along with L1, and MSE already compared in Khattak 2023 ("Self-regulating Prompts: Foundational Model Adaptation without Forgetting"). This isn’t mentioned in this paper and I think should be.


The main results shown in Table 1 compare ADAPT with various other studies of which Khattak 2023 is the closest in performance. However the figures are cherry picked from a larger set of results and misleading in that the chosen set (few-shot with k=16) is the ONLY setting in which ADAPT is actually better overall than Khattak 2023 - which isn’t mentioned in the main text (Appendix, Table 2 shows Khattak is better or the same with all k < 16). Moreover, the results with k=16 are close, and it would be appropriate to consider statistical significance.

**Requested Changes:**

Additional Experiments

- The Ablations subsection entitled “Maximum Context Length” isn’t an effective ablation - which should involve direct comparison between “with context pruning” and “without context pruning” to be convincing.

  I feel simple performance comparisons between your code, including angular knowledge distillation but with a set of fixed length prompts instead of context pruning - versus your code with  both angular knowledge distillation and context pruning, would be important in supporting claims concerning the effectiveness of the method in reducing computational overhead while preserving or enhancing performance.

- In addition, clearly noting the fixed prompt lengths utilised by other studies you compare to versus the average prompt length derived via the context pruning method would be informative and strengthen claims made. As things stand I wonder whether the prompt lengths used by some of your close comparisons are actually shorter than the average prompt length produced by context pruning.


Clarifications

- Claims such as “average test accuracy achieves the new state of the art” and “pruning greatly reduces the computational overhead” aren’t currently supported for reasons given above and should be re-worded or removed.

- Where different hyperparameters etc are used than those used by comparisons (eg. Khattak trains for 50 epochs and this paper trains for 100), such differences should be noted.


Minor clarification suggestions

- It isn’t clear whether the saliency calculation (equation 6) involves e.g. summing the three terms together or whether you tried each individually and settled on Snip in isolation.
- Adding a legend to Figure 1 distinguishing continuous from discrete token icons would reduce parsing time a bit.
- There are a few misspellings and grammatical errors.

---

> ### Author Response · Authors · 2025-11-22
>
> We thank the reviewer for constructive comments. We appreciate the time and effort devoted to evaluating our manuscript. Below is the one-to-one response:
>
> **Claim Made in the Submission**
>
> We want to make a clarification that we did mention that angular KD was inspired by Khattak et al. (2023):
>
> > The idea is inspired by the relational knowledge distillation (Park 2019) that proposes to learn relations of data samples and PromptSRC (Khattak 2023) that proposes to learn the text and image embeddings.
>
> Please refer to Subsection **Knowledge Distillation**.
>
> The reviewer commented that the main performance boost is through the knowledge distillation. The reviewer is comparing the Adapt approach without KD with PromptSRC. We want to emphasize that PromptSRC does use KD.
>
> The Adapt method is essentially a bi-level optimization with a lower level to optimize inserted soft prompts and an upper level to optimize the prompt configuration (i.e., context length). When the dataset size is very limited, it becomes difficult to be well-converged. Hence, when the number of shots is very small, the performance is worse than the best baseline.
>
> **Additional Experiment 1**
>
> We compared the performance with pruning to that without pruning. The result is shown in Appendix Figure 13 (b). We found that without pruning, there is performance degradation.
>
> Besides, in Figure 5, the red horizontal line indicates the performance of $T_{\rm effective} = 128$. In the non-pruning region where $T_{\rm effective} < T_{\rm target}$, two data points where $T_{\rm effective} = 240$ and $T_{\rm effective} = 192$ show worse performance despite that the total context length $T_{\rm effective}$ is larger.
>
> **Additional Experiment 2**
>
> The total context length in promptSRC is 72 while that for the Adapt method is 256. When increasing the context length of baseline methods, there will be performance degradation. For example, in the MaPLe baseline, a prompt depth of 9 is chosen. When increasing the prompt depth to 12, there is a performance drop [1].
>
> **Clarification 1**
>
> Following the reviewer's suggestion, we rephrased the claims. Please refer to the revised manuscript:
>
> > the average test accuracy achieves competitive performance
> > the pruning greatly reduces the computational overheads compared to unpruned prompt configuration
>
> **Clarification 2**
>
> We thank the reviewer for pointing out the differences in the hyperparameters. We reported the hyperparameters used in this study. Please refer to Section **Implementation Details**.
>
> > The total number of training epochs is 100.
>
> **Minor Clarification 1**
>
> We thank the reviewer for bringing it out. We only use Snip criterion. The result of using other criteria is shown in Appendix Figure 13 (c). To avoid potential confusion, we emphasized this result in the revised manuscript:
>
> > We use Snip to compute scores. The performance of the gradient norm and $l_2$-norm is reported in Appendix Figure 13 (c).
>
> **Minor Clarification 2**
>
> Following the reviewer's suggestion, we added the legend in Figure 1 to enhance the readability. Please refer to Figure 1 in the revised manuscript.
>
> **Minor Clarification 3**
>
> We have fixed the misspellings and grammatical errors. Please refer to the revised manuscript:
>
> > For the brevity -> For brevity
> > Fine-tuning pre-trained models in various downstream tasks is subjected to the forgetting issue -> Fine-tuning pre-trained models in various downstream tasks is subject to the forgetting issue
> > The performance of using fewer number of shows … -> The performance of using fewer number of shots …
> > Comparing to other prompting methods … -> Compared to other prompting methods…
> > context lengths for different layers can vary depending on the level of deviating from the optimal ->context lengths for different layers can vary depending on the level of deviation from the optimal
>
>
> [1] Khattak, Muhammad Uzair, et al. "Maple: Multi-modal prompt learning." Proceedings of the IEEE/CVF conference on computer vision and pattern recognition. 2023.

---

> ### Comment · Reviewer_hFh7 · 2025-11-24
>
> Thank you for the clarifications. My comments on your comments:
>
> #################
>
> We want to make a clarification that we did mention that angular KD was inspired by Khattak et al. (2023):
>
> The idea is inspired by the relational knowledge distillation (Park 2019) that proposes to learn relations of data samples and PromptSRC (Khattak 2023) that proposes to learn the text and image embeddings.
>
> REVIEWER: Thank you for clarifying that however I don't feel that a statement that Khattak 2023 "proposes to learn the text and image embeddings" preceded by a sentence that you "propose the angular knowledge distillation" really indicates that you are applying an existing technique.
>
> #################
>
> The reviewer commented that the main performance boost is through the knowledge distillation. The reviewer is comparing the Adapt approach without KD with PromptSRC. We want to emphasize that PromptSRC does use KD.
>
> REVIEWER: I am not confused about this. As mentioned, my comment is based on figure 4b which is comparing your setup with/without angular KD. Please note my other comments about the significance of your results versus PromptSRC also.
>
> #################
>
> The Adapt method is essentially a bi-level optimization with a lower level to optimize inserted soft prompts and an upper level to optimize the prompt configuration (i.e., context length). When the dataset size is very limited, it becomes difficult to be well-converged. Hence, when the number of shots is very small, the performance is worse than the best baseline.
>
> REVIEWER: I don't think that the sentence you currently have in section 4.2 that "performance with a fewer number of shots is in appendix..." is sufficient particularly given the paper still makes unqualified claims like "extensive experiments on 11 downstream datasets reveal the advantage of Adapt"
>
> #################
>
> Additional Experiment 1
> ...
> Additional Experiment 2
>
> The total context length in promptSRC is 72 while that for the Adapt method is 256. When increasing the context length of baseline methods, there will be performance degradation. For example, in the MaPLe baseline, a prompt depth of 9 is chosen. When increasing the prompt depth to 12, there is a performance drop [1].
>
> REVIEWER: Your "Additional Experiment 1" items compare different configurations of your context pruning code to each other which isn't an effective ablation. I accept that you have found configurations with smaller total pruned context length (128) than larger unpruned context lengths (192, 240) within your own code with differing parameters but I was seeking an ablation that completely removes your context pruning code and replaces it with "standard" fixed context code.
>
> Your "Additional Experiment 2" is closer to this and unfortunately indicates that a much smaller homogeneous fixed context (72) can perform approximately as well as Adapt (256) (82.87 vs 83.91). And I don't accept that increasing the prompt size would hurt performance in all fixed prompt cases and that your setup is the only one that doesn't!
>
> ######################
>
> Clarification 1
>
> the average test accuracy achieves competitive performance the pruning greatly reduces the computational overheads compared to unpruned prompt configuration
>
> REVIEWER: Per above eg PromptSRC fixed/unpruned context length 72!

---

> > ### Author Response · Authors · 2025-11-24
> >
> > We thank the reviewer for the reply. Currently, we are running the following experiments as the reviewer suggests:
> >
> > (1) Adapt with 256 context length without any pruning.
> > (2) PromptSRC with total context length of 256.
> >
> > We expect to finish experiments within 5 days and give a one-to-one response to all points the reviewer raised.

---

> > > ### Comment · Reviewer_hFh7 · 2025-11-24
> > >
> > > I wish you good luck with this assuming TMLR allows such extension, but if you want to convince me of the efficacy of context pruning please also run Adapt such that the final pruned context length is less than or equal to 75 to match the published PromptSRC setting.

---

> > > > ### Author Response · Authors · 2025-11-28
> > > >
> > > > We thank the reviewer for the reply. Below is the one-to-one response:
> > > >
> > > > **Comment 1**
> > > >
> > > > Following the reviewer's suggestion, we modified our statement:
> > > >
> > > > > Based on PromptSRC (Khattak 2023) where the l1 norm difference between embeddings of student model and teacher model is used as the knowledge distillation loss, and relational knowledge distillation where similarity is used as the knowledge distillation loss, we propose angular knowledge distillation, which aligns the cosine similarity between embeddings of the student model and the teacher model. In vision-language models, the prediction is based on cosine similarity between text and image embeddings. Angular knowledge distillation considers cosine similarity between the embeddings of the student model and the teacher model,  which preserves the prior knowledge in the teacher model.
> > > >
> > > > Please refer to the revised manuscript.
> > > >
> > > > **Comment 2**
> > > >
> > > > We believe that knowledge distillation is important in PEFT methods, as they can encounter forgetting issues in the fine-tuning process. Hence, our method does require knowledge distillation. As shown in Figure 4 (b) and Figure 13 (b), both knowledge distillation and heterogeneous prompts play an important role in the performance of the Adapt method.
> > > >
> > > > **Comment 3**
> > > >
> > > > Following the reviewer's suggestion, we modified our statement:
> > > >
> > > > > 16-shot experiments on 11 downstream datasets reveal the advantage of Adapt.
> > > >
> > > > > We notice that when the number of shots is less than 8, there is a degradation of the advantage of the Adapt method compared to baseline methods. This might be caused by poor convergence of bi-level optimization when the data are very limited.
> > > >
> > > > **Comment 4**
> > > >
> > > > In our Adapt method, when set to the non-pruning region, there is no pruning. It is essentially equal to removing the pruning code as the pruning is only done when $T_{\rm effective} > T_{\rm target}$. Please refer to Algorithm 1 Line 7 for the condition where pruning is applied.
> > > >
> > > > We examined the performance of the Adapt method with pruning and without pruning. A context length per layer of 12 is used for the unpruning setting, and the total context length is 288. We added the result in Figure 14 of the revised manuscript. Homogeneous prompts has an average accuracy of 83.91 while heterogeneous prompt has an average accuracy of 83.06. We also report the performance of the Adapt method when $T_{\rm target} = 72$ in Figure 14 of the revised manuscript.
> > > >
> > > > We examined the performance of PromptSRC when the complexity is closest to the Adapt method ($T_{\rm target} = 256$). For PromptSRC, we set the context length per layer to 12, the prompting depth to 12, and the total context length to 288. The performance comparison is shown in the following table:
> > > >
> > > > | Method | Caltech101 | DTD | EuroSAT | Aircraft | Food101 | Flowers | Pets | Cars | Sun | UCF | ImageNet | Average |
> > > > | ------- | ------ | ------- | ------ | ------- | ------ | ------- | ------ | ------ | ------ | ------- | ------ | ------ |
> > > > | Adapt | 96.57 | 74.07 | 93.53 | 58.30 | 86.67 | 98.43 | 92.67 | 85.97 | 86.43 | 77.17 | 73.20 | 83.91 |
> > > > | PromptSRC (288) | 96.37 | 74.70 | 93.17 | 51.73 | 87.63 | 98.07 | 94.00 | 84.83 | 86.31 | 77.17 | 72.13 | 83.28 |
> > > > | PromptSRC          | 96.04 | 72.75 | 92.48 | 50.86 | 87.44 | 97.62 | 93.81 | 83.79 | 86.43 | 77.24 | 73.12 |  82.87 |
> > > >
> > > > We notice that when total context length is 288, the performance of PromptSRC is still worse than Adapt ($T_{\rm target} = 256$).
> > > >
> > > > **Comment 5**
> > > >
> > > > We thank the reviewer for pointing it out. When we claim the computational overheads is reduced, we are comparing Adapt before pruning and Adapt after pruning. To avoid the potential misunderstanding, we added a clarification:
> > > >
> > > > > The comparison is based on Adapt before pruning and Adapt after pruning.
> > > >
> > > > Please refer to **Section 5 Discussion**.

---

### Review · Reviewer_hZcY · 2025-10-31

**Summary Of Contributions:**

- The method introduces an approach to automatically determine the context length based on depth, potentially leading to more efficient prompt designs.
- Adapt employs a saliency criterion to prune less important context tokens, aiming to reduce computational overhead while minimizing performance degradation.
- The paper suggests using angle knowledge distillation to mitigate forgetting during fine-tuning, encouraging the model to learn angular separations from the pre-trained model.
- Experiments on multiple downstream datasets show that Adapt achieves competitive performance, potentially outperforming existing methods.

**Audience:**

Yes

**Audience Explanation:**

surely there will be some individuals inerested in a potentially more efficient prompt design

**Broader Impact Concerns:**

no broader impact concerns

**Claims And Evidence:**

Yes

**Claims Explanation:**

basically all the claims made in the submission are supported by clear evidence.

**Requested Changes:**

Consider exploring the potential of extending this method to the NLP domain. The core concepts presented in this work don't seem inherently limited to vision tasks and might be applicable to natural language processing as well. This may strengthen the work.

---

> ### Author Response · Authors · 2025-11-21
>
> We thank the reviewer for constructive comments. We appreciate the time and effort devoted to evaluating our manuscript. Below is the one-to-one response:
>
> **Requested Change 1**
>
> Following the reviewer's suggestion, we applied the proposed method in the NLP domain. The performance comparison is shown below:
>
> | Method        | COPA # param | OCPA Acc | BoolQ # Param | BoolQ Acc | RTE # param | RTE ACC |
> | -------------- | ----------------- | ------------ | ------------------ | ------------ | --------------- | ----------- |
> | P-Tuning v2 | 0.787 M            | 78.00         | 1.968 M             | 75.02         | 0.985 M         | 78.17 |
> | Adapt           | 0.297 M            | 80.00         | 0.297 M             | 76.50         | 0.297 M         | 79.17 |
>
> We found that the Adapt method exhibits better performance compared to the baseline P-Tuning v2 [1].
>
> [1] Liu, Xiao, et al. "P-tuning: Prompt tuning can be comparable to fine-tuning across scales and tasks." Proceedings of the 60th Annual Meeting of the Association for Computational Linguistics (Volume 2: Short Papers). 2022.

---

> > ### Comment · Reviewer_hZcY · 2025-11-22
> >
> > Thanks for addressing my comments. I really appreciate you exploring the method in the NLP domain and showing those improved results. I think this paper is solid and would be a great fit for TMLR.

---

> > > ### Author Response · Authors · 2025-11-22
> > >
> > > We are grateful that the reviewer found our responses satisfactory. The comments have been highly valuable in improving the quality of the manuscript.
> > >
> > > We thank the reviewer again for the thoughtful evaluation!

---

### Review · Reviewer_qPK4 · 2025-11-04

**Summary Of Contributions:**

This paper proposes a novel, efficient parameter cue tuning method called ADAPT, which applies to pre-trained visual language models, such as CLIP. Its core contribution lies in introducing heterogeneous deep cue information, where the length of consecutive cue words can vary depending on the Transformer layerr.

This contrasts sharply with standard deep cue tuning methods, which use cue words of uniform (fixed) length across all depths. The authors hypothesize that different layers of a pre-trained model have varying degrees of deviation from their optimal weights for new tasks, thus requiring different degrees of adaptive adjustment.

ADAPT automatically learns these different cue word lengths, rather than manually designing them. The implementation is as follows:

Initially, all layers use large, fixed-length cue words.

During training stage, individual cue words are iteratively pruned (removed) based on a saliency criterion (specifically, a snip), which measures the contribution of each cue word to the loss.

This automatic pruning process ultimately generates a sparse cue structure with varying layer lengths, enabling adaptation to specific downstream tasks.

As another contribution: this paper also introduces a novel angular knowledge distillation (KD) loss function. This loss function aims to prevent catastrophic forgetting by forcing the student model to maintain the angular separation (cosine similarity) between its class embeddings and instance embeddings, as well as the original pre-trained teacher model. The authors argue that this approach outperforms the traditional logit-based KD loss function because the zero-sample logit value of the teacher model is often not optimal.

**Additional Comments:**

N/A

**Audience:**

Yes

**Audience Explanation:**

This paper focuses on PEFT of VLMs, which is a highly significant and active area of research within the machine learning community, particularly relevant to TMLR's audience.

The core innovation of the paper lies in its proposal to integrate concepts from network pruning with prompt learning to automate the design of prompt structures. This constitutes a novel contribution to the field. Researchers and practitioners engaged in model adaptation, efficient learning, and vision-language models will likely find this state-of-the-art method and its substantial insights—including the advantages of pruning compared to training a small prompt from the ground up, as well as the effectiveness of angular knowledge distillation over traditional logit knowledge distillation—particularly compelling.

**Broader Impact Concerns:**

The paper does not currently include a Broader Impact Statement. As a work on parameter-efficient model adaptation, it falls into the category of general-purpose machine learning research. It does not introduce any ethical risks beyond those already inherent in large-scale vision-language models (e.g., potential for bias amplification from downstream datasets, potential for misuse in surveillance or deepfake generation).

In fact, PEFT methods like this one can be seen as having a positive impact by lowering the computational (and thus environmental) cost of fine-tuning, making adaptation more accessible.

I do not see any specific, unaddressed broader impact concerns that would require remediation. However, I would encourage the authors to add a standard, brief Broader Impact Statement acknowledging these general points, as is common practice.

**Claims And Evidence:**

Yes

**Claims Explanation:**

The paper's main claims are well-supported by extensive experiments and ablations.

**Claim (SOTA Performance)**: The central claim that ADAPT outperforms existing methods is convincingly demonstrated in Table 1. The proposed method achieves the highest average accuracy (83.91%) across 11 datasets, surpassing all listed baselines in the 16-shot setting.

**Claim (Benefit of Angular KD)**: The claim that Angular Knowledge Distillation is a key contributor to this performance is clearly validated by the ablation study in Figure 4. Adding Angular KD provides a significant boost to the average test accuracy (from ~81.9% to ~83.9%), and the t-SNE plots qualitatively support the claim that it helps create better-separated clusters.

**Claim (Benefit of Pruning Process)**: The claim that the process of pruning from a large search space is beneficial (and not just the final reduced parameter count) is strongly supported by Figure 5. This ablation shows that a model pruned down to a target length (e.g., T_effective=128) from a large initial space (xi=18) significantly outperforms models that start with a smaller, fixed prompt length (e.g., T_effective=192 in the shaded area), even when the latter has more parameters. This is a crucial and convincing piece of evidence.

**Requested Changes:**

I have some questions and concerns need to ask the authors.

**Concerns:**

**Clarify or Justify Saliency Metrics (Eq 6)**: Equation 6 lists three different saliency criteria: Snip, gradient norm, and $l_2$-norm. However, the caption for Table 1 states, "Adapt uses Snip," and no ablation or comparison between these three criteria is ever presented. This is misleading, as it claims three potential methods but only provides evidence for one of them. Required Action: The authors must either (a) provide an ablation study comparing the performance of all three saliency metrics or (b) (simpler) remove the "gradient norm" and "$l_2$-norm" from Equation 6 and state clearly that Snip was chosen, perhaps leaving the others as a brief mention for future work.

**Directly Compare Angular KD vs. Standard KD**: The paper argues that standard (logit-based) KD is suboptimal but never provides evidence to support this claim. Moreover, one more question: what is the difference between ADAPT and [1]?



**Exsiting Methods**: When a paper transitions from "PEFT" to "prompting methods," recommended authors add [2] to further discussion. The ADAPT paper mentions that CoCoOp is "conditioned on the input image." Following this line of thought, [3] further explores how to generate specific cues for each visual instance. However, these methods primarily focus on instance adaptation of cue content. In contrast, the authors' ADAPT method addresses a complementary problem: automatically learning an optimal, heterogeneous cue structure (i.e., different lengths at different depths), optimized for the entire downstream task rather than for a single instance. It is recommended to include some discussion or comparison.

**Question 1**: Pattern Analysis Regarding "Inter-Layer Deviation": The core motivation of the paper is that different layers of the pre-trained model exhibit 'different levels of deviation' for downstream tasks. You obtained a heterogeneous cue structure through automatic pruning, which validates the heterogeneity of the results. However, what is the pattern of this heterogeneity? For example, did you observe a consistent pattern across all 11 datasets? For instance, is (a) the shallower layers (e.g., layers 1-4) always pruned more sparsely, while deeper layers (e.g., layers 9-12) retain more cues? Or is (b) the pattern completely random and highly dependent on a specific dataset? If a consistent pattern exists (e.g., always pruning shallow layers), does this mean your method can be approximated by a simpler, non-automated 'heuristic' (e.g., adding cues only in the last six layers)? Quantitative analysis of the pruned structure (not just visualization) seems necessary to validate your core motivation.


**Question 2**: Applicability to Pure Vision Models: The paper's method achieved an SOTA score on CLIP. However, can and how can the ADAPT method be applied to purely visual models (e.g., a standard ViT pre-trained on ImageNet-21k)? Specifically, your Angular KD loss (Eq 10) contains a L_ang^text term that relies on the text encoder to compute class embeddings. In a purely visual model, without a text encoder, will your knowledge distillation mechanism completely fail? Alternatively, can ADAPT outperform baselines like VPT on purely visual models simply by relying on pruning, without Angular KD?

These 2 questions just some of my questions and will not affect my rating of this paper. This is a very strong paper in prompt tuning. As I am also a researcher in the field of visual prompt tuning, I welcome any points of discussion if the authors believe I have misunderstood something.

*References*

[1] "E^2VPT: An Effective and Efficient Approach for Visual Prompt Tuning" ICCV 2023

[2] "Prompt-based Adaptation in Large-scale Vision Models: A Survey" arXiv 2025

[3] "Visual Instance-aware Prompt Tuning" ACM MM 2025

---

> ### Author Response · Authors · 2025-11-21
>
> We thank the reviewer for constructive comments. We appreciate the time and effort devoted to evaluating our manuscript. Below is the one-to-one response:
>
> **Concern 1**
>
> We reported the result of using the gradient norm and $l_2$-norm in Appendix Figure 13 (c). We thank the reviewer for pointing it out. To avoid potential confusion, we emphasized this result in the updated manuscript:
>
> > We use Snip to compute scores. The performance of gradient norm and $l_2$-norm is reported in Appendix Figure 13 (c).
>
> **Concern 2**
>
> Following the reviewer's suggestion, the performance comparison between angular KD and standard KD using the Adapt method is listed in the following table:
>
> | Method                    | Average Performance |
> | ---------------------- | ------------------------- |
> | Adapt w/o KD          | 81.90                          |
> | Adapt Standard KD | 82.63                          |
> | Adapt Angular KD   | 83.91                          |
>
> We listed the differences between the Adapt method and reference [1] that the reviewer mentioned:
>
> - In addition to having inserted soft prompts trainable, [1] also has multihead self-attention (MSA) trainable. Adapt is purely soft prompting-based methods.
> - [1] uses the pruning + rewinding strategy while Adapt prunes prompts progressively during the fine-tuning process.
> - Adapt controls the computational complexity by setting $T_{\rm target}$ while [1] does not have such a computational budget.
> - Adapt uses angular knowledge distillation to avoid the forgetting issue while [1] does not use knowledge distillation.
>
> **Concern 3**
>
> Following the reviewer's suggestion, we added recommended references in the discussion. Please refer to the revised manuscript.
>
> > Prompt-based adaptation has been applied in various areas, including foundational CV tasks, constrained learning, trustworthy AI, foundational analysis, etc. [2].
> > ViAPT proposes instance-aware prompts to balance dataset-level and instance-level knowledge [3].
>
> **Question 1**
>
> We do not observe a uniform pattern across datasets. We added the analysis of context lengths in Appendix A7 of the revised manuscript. Please refer to Appendix Figure 11.
>
> **Question 2**
>
> Without using angular KD, Adapt can be applied to pure visual models by inserting soft prompts to the input to every transformer block followed by a pruning process.
>
> Angular KD is designed to align the cosine similarity between image embedding and text embeddings with the teacher model. In the multimodal models, cosine similarity is commonly used to align different modalities. This does not apply to a single-modal model.
>
> Without using angular KD, the average accuracy over 11 datasets for the Adapt method is 81.90. The performance of VPT-Deep is 79.38 and that of VPT-Shallow is 73.18 (please refer to Table 1). The adapt method still exhibits better performance even without relying on angular KD.
>
> **Broader Impact**
>
> Following the reviewer's suggestion, we added the Section "Broader Impact". Please refer to the revised manuscript.
>
> > We propose an automatic way to determine a heterogeneous prompt, which can be applied to fine-tune foundational models in various downstream tasks efficiently. Different from existing manually designed prompting approaches that use a homogeneous prompt, heterogeneous prompts improve efficiency by pruning unimportant context tokens.

---

> ### Comment · Reviewer_qPK4 · 2025-11-22
> **Respond to authors**
>
> Thank the authors for the detailed reply. I am very satisfied with your response. This is already a very good work, insight a lot, so I think it  can be published in TMLR. Good luck.

---

> > ### Author Response · Authors · 2025-11-22
> >
> > We are grateful that the reviewer found our responses satisfactory and the work insightful. The comments have been highly valuable in improving the clarity and quality of the manuscript.
> >
> > We thank the reviewer again for the thoughtful evaluation!

---

### Review · Reviewer_Mc6r · 2025-11-05

**Summary Of Contributions:**

The paper presents ADAPT, a method for parameter-efficient fine-tuning (PEFT) of Vision-Language Models (VLMs) that addresses two key challenges: prompt redundancy and catastrophic forgetting. Instead of relying on fixed-length prompts, ADAPT introduces heterogeneous, layer-wise context lengths through an iterative pruning process. This process uses a saliency criterion to estimate token importance and dynamically remove less informative context tokens. To stabilize fine-tuning, the method incorporates Angular Knowledge Distillation (AKD), which regularizes training by aligning the angular relationships between class and instance embeddings in the fine-tuned model and those of the frozen teacher. Across 11 benchmark datasets, ADAPT shows consistent improvements over fixed-length prompt-tuning baselines, indicating more efficient and stable adaptation.

The main novelty of ADAPT lies in its architectural design and stability mechanism. The use of adaptive, layer-wise context lengths is both original and well-motivated, as deeper layers naturally benefit from different contextual information than shallower ones. This approach provides an elegant way to address prompt redundancy through saliency-based pruning. The addition of AKD is also well-justified, offering a principled method to mitigate catastrophic forgetting, which is a common challenge in PEFT methods. Overall, the framework is clearly structured, and the motivation for pruning-based efficiency is easy to follow.

However, I have several (perhaps minor) concerns regarding generalization and practical efficiency. The experiments are limited to the CLIP architecture, leaving open the question of how well ADAPT would generalize to other VLMs with different attention mechanisms or training objectives. More importantly, while the paper claims improved efficiency through parameter reduction, it does not provide direct evidence in terms of runtime performance, such as wall-clock training time, inference latency, or FLOPs. Without this information, it is difficult to determine whether the pruning overhead is truly offset by the parameter savings. The use of heterogeneous context lengths may also introduce implementation challenges, particularly on hardware optimized for fixed tensor sizes. It is not clear how the dynamic context representation affects batching efficiency and parallelization on GPUs or TPUs.

**Audience:**

Yes

**Audience Explanation:**

I think this paper makes a meaningful contribution to ongoing work in parameter-efficient fine-tuning, addressing the key trade-off between model adaptability and resource use. The idea of adaptive context length is an interesting and potentially influential architectural innovation for researchers working on prompt design and PEFT methods.

**Claims And Evidence:**

Yes

**Claims Explanation:**

The improvements in accuracy and parameter efficiency over the baselines are well-supported by the results in Table 1. The ablation studies also indicate that both adaptive context pruning and AKD contribute meaningfully to the performance gains. However, the efficiency claim currently rests only on the reduction in trainable parameters, without accompanying evidence of lower computational cost (e.g., wall-clock time or FLOPs).

**Requested Changes:**

The paper’s main efficiency claim should be supported with concrete runtime evidence. Please include quantitative comparisons of both training wall-clock time and inference latency (in milliseconds or seconds per batch) between ADAPT and a comparable strong baseline, e.g., Khattak et al. (2023) or standard fixed-length deep prompt tuning. This will help demonstrate that any pruning overhead is indeed offset by the reduction in parameter usage.

In addition, it would be valuable to include an analysis or ablation quantifying the computational cost of the iterative saliency estimation and pruning steps, ideally expressed as a percentage of total training time. This would help clarify any hidden overhead involved in constructing the heterogeneous context structure.

To better assess ADAPT’s robustness, consider evaluating it on at least one substantially different VLM architecture (for instance, one based on ALIGN or CoCa). This would strengthen the generality claims of the approach.

The paper should also explain why heterogeneous context lengths are necessary within the fixed-tensor-size design typical of modern VLM attention layers, and discuss any resulting implications. For example, does this setup require dynamic padding or specialized masking? If so, how is training efficiency maintained under these conditions?

There are a few minor typographical errors that should be corrected (non-comprehensive):

- “adaptive prompt tuing” → “adaptive prompt tuning” (Section 1)

- “DescribableTectures” → “Describable Textures” (dataset list in Section 4.1)

- “improves the the best-known performance” → remove the duplicated “the” (Section 4.2)

- “optimation objective” → “optimization objective” (Section 3.2)

- “number of shows is 16” → “number of shots is 16” (Table 1 caption)

- "using fewer number of shows" → “using fewer number of shots” (Section 4.2)

---

> ### Author Response · Authors · 2025-11-21
>
> We thank the reviewer for constructive comments. We appreciate the time and effort devoted to evaluating our manuscript. Below is the one-to-one response:
>
> **Requested change 1**
>
> We want to make a clarification that efficiency is achieved by pruning unimportant context tokens, which reduces computational complexity. The efficiency claim is based on the comparison between unpruned context tokens and pruned context tokens. Following the reviewer's suggestion, we examined the wall-clock comparison between PromptSRC (Khattak et al 2023) and proposed Adapt approach. We use the same hyperparameters: the context length for each transformer block is 6 and batch size is 32. For the Adapt method, we use $T_{\rm target} = 128$. The average clock time over 11 datasets is:
>
> | Method        | Train                | Inference |
> | -------------- | ---------------- | ---------- |
> | PromptSRC | 1.1 min/epoch | 1.47 min |
> | Adapt          | 1.2 min/epoch  | 1.48 min |
>
> We found that the average clock time for two methods is similar.
>
> **Requested change 2**
>
> We believe that the computational costs for computing the importance score and pruning is negligible compared to training costs. We analyzed the computational costs:
>
> **Cost of computing the importance score**: we used Snip to compute the importance score, which requires the computation of the gradient to update the prompt and the magnitude of the prompt. The gradient computation is already finished in the back propagation of updating soft prompts. The cost of computing the magnitude of the soft prompt is essentially the cost of computing $l_2$ norm.
>
> **The cost of purning context tokens**: the pruning process is merely element-wise multiplication between mask and prompts.
>
> Hence, we believe the computational costs for computing the importance score is negligible. Following the reviewer's suggestion, we examined the running time between Adapt and Adapt without pruning. The same complexity is ensured for the Adapt and Adapt without pruning. The clock time comparison is:
>
> | Method                   | Train                | Inference      |
> | ---------------------- | ---------------- | -------------- |
> | Adapt (No pruning) | 1.1 min/epoch | 1.47 min      |
> | Adapt                      | 1.2 min/epoch | 1.48 min      |
>
> **Requested change 3**
>
> We tried to test the performance of the Adapt method on different VLLMs. To our best knowledge, we found ALIGN [1] was not an open source model. Coca [2] does not release pre-trained weights. Hence, we choose a different VLLM: SigLIP [3]. The performance comparison with PromptSRC is:
>
> | Method | Caltech101 | DTD | EuroSAT | Aircraft | Food101 | Flowers | Pets | Cars | Sun | UCF | ImageNet | Avg |
> | -------------- | ------ | ------- | ------ | ------- | ------ | ------- | ------ | ------ | ------- | ------ | ------- | ------- |
> | PromptSRC | 97.20 | 72.30 | 92.33 | 51.63 | 90.13 | 95.31 | 94.81 | 90.12 | 82.67 | 77.83 | 78.36 | 83.88 |
> | Adapt           | 96.50 | 74.53 | 94.71 | 58.43 | 88.53 | 95.67 | 93.92 | 90.37 | 82.43 | 77.63 | 78.42 | 84.65 |
>
> We found that our method still exhibits better performance than the baseline PromptSRC.
>
> **Requested change 4**
>
> We found that the heterogeneous prompting approach leads to better performance compared to the homogeneous prompting approach. The performance comparison is shown in Appendix Figure 13 (b).
>
> Besides, in Figure 5, the red horizontal line indicates the performance of $T_{\rm effective} = 128$. In the non-pruning region where $T_{\rm effective} < T_{\rm target}$, two data points where $T_{\rm effective} = 240$ and $T_{\rm effective} = 192$ show worse performance despite that the total context length $T_{\rm effective}$ is larger.
>
> The heterogeneous prompting does not require dynamic padding or specialized masking. By utilizing the cross attention, the outputs of transformer blocks have the same sequence length as the inputs of transformer blocks (i.e., the sequence length without inserting soft prompts).
>
> We modified the typos pointed out by the reviewer. Please refer to the revised manuscript.
>
> [1] https://research.google/blog/align-scaling-up-visual-and-vision-language-representation-learning-with-noisy-text-supervision
>
> [2] https://github.com/lucidrains/CoCa-pytorch
>
> [3] Zhai, Xiaohua, et al. "Sigmoid loss for language image pre-training." Proceedings of the IEEE/CVF international conference on computer vision. 2023.

---

> > ### Comment · Reviewer_Mc6r · 2025-12-04
> >
> > Thank to the authors for the detailed response. I also see that initial concerns by the other reviewers were largely addressed, so I do not see any issues from my side.

---

> > > ### Author Response · Authors · 2025-12-04
> > >
> > > We are grateful that the reviewer found our responses satisfactory and the work insightful. The comments have been highly valuable in improving the clarity and quality of the manuscript.
> > >
> > > We thank the reviewer again for the thoughtful evaluation!

---

### Decision · Action_Editor_aMmS · 2025-12-23

**Recommendation:** Reject

**Additional Comments:**

The resubmission should carefully address the concerns of Reviewer hFh7, who remained unconvinced even after the 2nd revision, especially relative to Khattak 2023. Below are the points raised by this reviewer (copy-pasted):

The primary focus in the paper is the "context pruning method" based on saliency. Unfortunately, the results (Figure 4b) suggest that the main driver of improvement is actually the added regularisation term based on cosine similarity ("angular knowledge distillation"). This "angular knowledge distillation" approach was one of several similar methods along with L1, and MSE already compared in Khattak 2023 ("Self-regulating Prompts: Foundational Model Adaptation without Forgetting"). The way the Adapt paper is written still implies that "angular knowledge distillation" is their innovation when it is clearly one of the methods evaluated in Khattak 2023 (see Khattak table 5 and associated text). Moreover, since most of the improvement seems to come from the "angular knowledge distillation", doubts are raised about the efficacy of the main innovation proposed, the "context pruning method".

The main results shown in Table 1 compare ADAPT with various other studies of which Khattak 2023 is the closest in performance. However the figures are cherry picked from a larger set of results and misleading in that the chosen set (few-shot with k=16) is the ONLY setting in which ADAPT is actually better overall than Khattak 2023. The authors do now acknowledge this in the main text, which is an improvement over prior versions. Unfortunately, even the results with k=16 are close, and it would be appropriate to consider statistical significance which is not done after repeated requests.

The Ablations subsection entitled “Maximum Context Length”, which might have been expected to quantify the effectiveness of the "context pruning method" isn’t an effective ablation. This section should involve direct comparison between “with context pruning” and “without context pruning” to be convincing, but instead compares differing versions of their "context pruning method" code to each other (and do indeed find configurations of their code where no pruning with larger context underperforms pruning with smaller resulting context). Noting that the authors report Adapt results with total context length 256 and Khattak reported results only use a total context length of 75, it is reasonable to question to efficacy of their context pruning method in actually reducing the amount of needed compute to produce a particular result compared to the current standard practice of using a fixed context length. The authors then ran further experiments for Khattak at context length 288 and their own Adapt method at 75 and put results in an appendix rather than the main text. Unfortunately the results are inconclusive with Adapt being slightly stronger at 256/288 (Adapt 83.91 vs Khattak 83.28) but slightly weaker at 75 (Adapt 82.75 vs Khattak 82.87). Overall I think the strongest claim that could be made from the data presented is that there is weak evidence supporting the notion that context pruning might confer a small benefit versus current baselines when the target context size is larger than what current baselines have considered (i.e. >=256) but this would need further analysis and proper consideration of statistical significance.

**Audience:**

Yes

**Audience Explanation:**

Parameter-efficient fine-tuning of vision-language models is an active research area, and the question of whether heterogeneous context lengths outperform fixed-length prompts is a reasonable one to explore. Even if the current evidence is inconclusive, the idea of applying saliency-based pruning to prompt design could inspire follow-up work. Researchers working on prompt tuning and VLM adaptation would likely find the empirical comparisons and ablations informative.

**Claims And Evidence:**

No

**Claims Explanation:**

The paper's central claim—that adaptive context pruning improves efficiency and performance—is undermined by the results that: when context lengths are matched (75 tokens), ADAPT (82.75) actually underperforms Khattak 2023 (82.87), and only wins when given a larger starting budget (256). Figure 4b suggests Angular KD drives most gains. Additionally, the k=16 setting is the only few-shot configuration where ADAPT beats the closest baseline. The close margins also warrant statistical significance testing, which is absent. The evidence is accurate in what it reports but not convincing that the core proposed mechanism provides substantial benefit beyond existing approaches.

**Resubmission Of Major Revision:**

The authors may consider submitting a major revision at a later time.